# Single Layer Predictive Normalized Maximum Likelihood for Out-of-Distribution Detection

**Koby Bibas**
School of Electrical Engineering
Tel Aviv University
kobybibas@gmail.com

**Meir Feder**
School of Electrical Engineering
Tel Aviv University
meir@eng.tau.ac.il

**Tal Hassner**
Facebook AI
talhassner@gmail.com

## Abstract

Detecting out-of-distribution (OOD) samples is vital for developing machine learning based models for critical safety systems. Common approaches for OOD detection assume access to some OOD samples during training which may not be available in a real-life scenario. Instead, we utilize the *predictive normalized maximum likelihood* (pNML) learner, in which no assumptions are made on the tested input. We derive an explicit expression of the pNML and its generalization error, denoted as the *regret*, for a single layer neural network (NN). We show that this learner generalizes well when (i) the test vector resides in a subspace spanned by the eigenvectors associated with the large eigenvalues of the empirical correlation matrix of the training data, or (ii) the test sample is far from the decision boundary. Furthermore, we describe how to efficiently apply the derived pNML regret to any pretrained deep NN, by employing the explicit pNML for the last layer, followed by the softmax function. Applying the derived regret to deep NN requires neither additional tunable parameters nor extra data. We extensively evaluate our approach on 74 OOD detection benchmarks using DenseNet-100, ResNet-34, and WideResNet-40 models trained with CIFAR-100, CIFAR-10, SVHN, and ImageNet-30 showing a significant improvement of up to 15.6% over recent leading methods.

## 1 Introduction

An important concern that limits the adoption of deep neural networks (DNN) in critical safety systems is how to assess our confidence in their predictions, i.e, quantifying their *generalization capability* (Kaufman et al., 2019; Willers et al., 2020). Take, for instance, a machine learning model for medical diagnosis (Bibas et al., 2021). It may produce (wrong) diagnoses in the presence of test inputs that are different from the training set rather than flagging them for human intervention (Singh et al., 2021). Detecting such unexpected inputs had been formulated as the out-of-distribution (OOD) detection task (Hendrycks and Gimpel, 2017), as flagging test inputs that lie outside the training classes, i.e., are not *in-distribution* (IND).

Previous learning methods that designed to offer such generalization measures, include VC-dimension (Zhong et al., 2017; Vapnik and Chervonenkis, 2015) and norm based bounds (Neyshabur et al., 2018; Bartlett et al., 2017). As a whole, these methods characterized the generalization ability based on the properties of the parameters. However, they do not consider the test sample that is presented to the model (Jiang et al., 2020), which makes them useless for OOD detection. Other approaches build heuristics over the *empirical risk minimization* (ERM) learner, by post-processing

the model output (Sastry and Oore, 2020) or modifying the training process (Papadopoulos et al., 2021; Vyas et al., 2018). Regardless of the approach, these methods choose the learner that minimizes the loss over the *training set*. This may lead to a large generalization error because the ERM estimate may be wrong on unexpected inputs; especially with large models such as DNN (Belkin et al., 2019).

To produce a useful generalization measure, we exploit the *individual setting* framework (Merhav and Feder, 1998). In the individual setting, there is no assumption about how the training and the test data are generated, nor about their probabilistic relationship. Moreover, the relationship between the labels and data can be deterministic and may therefore be determined by an adversary. The generalization error in this setting is often referred to as the *regret* (Merhav and Feder, 1998). This regret is defined by the log-loss difference between a learner and the *genie*: a learner that knows the specific test label, yet is constrained to use an explanation from a set of possible models. The individual setting is the most general framework and so the result holds for a wide range of scenarios. Specifically, the result holds to OOD detection where the distribution of the OOD inputs is unknown.

The pNML learner (Fogel and Feder, 2018) was proposed as the min-max solution of the regret, where the minimum is over the learner choice and the maximum is for any possible test label value. Intuitively, the pNML assigns a probability for a potential outcome as follows: Add the test sample to the training set with an arbitrary label, find the ERM solution of this new set, and take the probability it gives to the assumed label. Follow this procedure for every label and normalize to get a valid probability assignment. The pNML was developed before for linear regression (Bibas et al., 2019b) and was evaluated empirically for DNN (Fu and Levine, 2021; Bibas et al., 2019a).

We derive an analytical solution of the pNML learner and its generalization error (the regret) for a single layer NN. We analyze the derived regret and show it obtains low values when the test input either (i) lies in a subspace spanned by the eigenvectors associated with the large eigenvalues of the training data empirical correlation matrix or (ii) is located far from the decision boundary. Crucially, although our analysis focuses on a single layer NN, our results are applicable to the last layer of DNNs *without changing the network architecture or the training process*: We treat the pretrained DNN as a feature extractor with the last layer as a single layer NN classifier. We can therefore show the usage of the pNML regret as a confidence score for the OOD detection task.

To summarize, we make the following contributions.

1. **Analytical derivation of the pNML regret.** We derive an analytical expression of the pNML regret, which is associated with the generalization error, for a single layer NN.

2. **Analyzing the pNML regret.** We explore the pNML regret characteristics as a function of the test sample data, training data, and the corresponding ERM prediction. We provide a visualization on low dimensional data and demonstrate the situations in which the pNML regret is low and the prediction can be trusted.

3. **DNN adaptation.** We propose an adaptation of the derived pNML regret to *any* pretrained DNN that uses the softmax function with neither additional parameters nor extra data.

Applying our pNML regret to a pretrained DNN does not require additional data, it is efficient and can be easily implemented. The derived regret is theoretically justified for OOD detection since it is the individual setting solution for which we do not require any knowledge on the test input distribution. Our evaluation includes 74 IND-OOD detection benchmarks using DenseNet-BC-100, ResNet-34, and WideResNet-40 trained with CIFAR-100, CIFAR-10, SVHN, and ImageNet-30. Our approach outperforms leading methods in nearly all 74 OOD detection benchmarks up to a remarkable $+15.2\%$

## 2 Related work

**OOD detection.** Many recent work use additional OOD set in training (Malinin and Gales, 2018; Hendrycks et al., 2019a; Nandy et al., 2020; Mohseni et al., 2020). A well-known method for OOD detection, named *ODIN* (Liang et al., 2018), manipulates the input image based on the gradient of the loss function. This input manipulation increases the margin between the maximum probability of images with known classes and images with unknown objects. However, the manipulation strength is defined by OOD data, which might be unavailable or different from the OOD samples in inference.

Different approaches propose to manipulate the loss function in the training phase. Lee et al. (2018) proposed to use the Mahalanobis distance between the test input representations and the class conditional distribution in the intermediate layers to train a logistic regression classifier to determine if the input sample is OOD. The *Energy* method (Liu et al., 2020) adds a cost function to the training phase to shape the energy surface explicitly for OOD detection. Papadopoulos et al. (2021) suggested the *outlier exposure with confidence control* (OECC) method in which a regularization term is added to the loss function such that the model produces a uniform distribution for OOD samples.

All mentioned methods require manipulating the DNN architecture or adding additional data. The *Baseline* method (Hendrycks and Gimpel, 2017) is one of the earliest approaches designed to identify whether a test input is IND and does not require changing the pretrained model. This method uses the maximum probability of the DNN output as the OOD score. The *Gram* method (Sastry and Oore, 2020) detects OODs based on feature representations obtained in intermediate layers. Given a test sample, the Gram matrices of the test sample features are compared with those of the training samples known to belong to the estimated class of the test sample. However, this method requires IND validation set and does not have a theoretical motivation.

Other work suggested using Bayesian techniques to estimate the prediction confidence (Gal and Ghahramani, 2016; Lakshminarayanan et al., 2017; van Amersfoort et al., 2020). However, the Bayesian techniques add extensive compute overhead to the prediction.

**The pNML learner.** The pNML learner is the min-max solution of the supervised batch learning in the individual setting (Fogel and Feder, 2018). For sequential prediction it is termed the conditional normalized maximum likelihood (Rissanen and Roos, 2007; Roos and Rissanen, 2008).

Several methods deal with obtaining the pNML learner for different hypothesis sets. Bibas et al. (2019b) and Bibas and Feder (2021) showed the pNML solution for linear regression. Rosas et al. (2020) proposed an NML based decision strategy for supervised classification problems and showed it attains heuristic PAC learning. Fu and Levine (2021) used the pNML for model optimization based on learning a density function by discretizing the space and fitting a distinct model for each value.

For the DNN hypothesis set, Bibas et al. (2019a) estimated the pNML distribution with DNN by fine-tuning the last layers of the network for every test input and label combination. This approach is computationally expensive since training is needed for every test input. Zhou and Levine (2020) suggested a way to accelerate the pNML computation in DNN by using approximate Bayesian inference techniques to produce a tractable approximation to the pNML. Pesso et al. (2021) used the pNML with adversarial target attack as a defense mechanism against adversarial perturbation.

## 3 Notation and preliminaries

In the supervised machine learning scenario, a training set consisting of $N$ pairs of examples is given

$$\mathcal{D}_N = \{(x_n, y_n)\}_{n=1}^N, \quad x_n \in \mathcal{X}, \quad y_n \in \mathcal{Y}, \tag{1}$$

where $x_n$ is the $n$-th data instance and $y_n$ is its corresponding label. The goal of a learner is to predict the unknown test label $y$ given a new test data $x$ by assigning probability distribution $q(\cdot|x)$ to the unknown label. The performance is evaluated using the log-loss function

$$\ell(q; x, y) = -\log q(y|x). \tag{2}$$

For the problem to be well-posed, we must make further assumptions on the class of possible models or *hypothesis set* that is used to find the relation between $x$ and $y$. Denote $\Theta$ as a general index set, this class is a set of conditional probability distributions

$$P_\Theta = \{p_\theta(y|x), \ \theta \in \Theta\}. \tag{3}$$

The ERM is the learner from the hypothesis set that attains the minimal log-loss on the training set.

**The individual setting.** An additional assumption required to solve the problem is related to how the data and the labels are generated. In this work we consider the individual setting (Merhav and Feder, 1998; Fogel and Feder, 2018; Bibas et al., 2019b,a), where the data and labels, both in the training and test, are specific individual quantities: We do not assume any probabilistic relationship between them, the labels may even be assigned in an adversarial manner.

**The genie.** In the individual setting the goal is to compete with a reference learner, a genie, that has to following properties: (i) knows the test label value, (ii) is restricted to use a model from the given hypotheses set $P_\Theta$, and (iii) does not know which of the samples is the test. This reference learner then chooses a model that attains the minimum loss over the training set and the test sample

$$\hat{\theta}(\mathcal{D}_N; x, y) = \arg\min_{\theta \in \Theta} \left[ \ell\left(p_\theta; x, y\right) + \sum_{n=1}^{N} \ell\left(p_\theta; x_n, y_n\right) \right]. \tag{4}$$

The regret is the log-loss difference between a learner $q$ and this genie:

$$R(q; \mathcal{D}_N; x, y) = -\log q(y|x) - \left[ -\log p_{\hat{\theta}(\mathcal{D}_N; x, y)}(y|x) \right]. \tag{5}$$

**Theorem 1** (Fogel and Feder (2018))**.** *The universal learner, denoted as the pNML, minimizes the regret for the worst case test label*

$$\Gamma = R^*(\mathcal{D}_N, x) = \min_q \max_{y \in \mathcal{Y}} R(q; \mathcal{D}_N; x, y). \tag{6}$$

*The pNML probability assignment and regret are*

$$q_{pNML}(y|x) = \frac{p_{\hat{\theta}(\mathcal{D}_N; x, y)}(y|x)}{\sum_{y' \in \mathcal{Y}} p_{\hat{\theta}(\mathcal{D}_N; x, y')}(y'|x)}, \quad \Gamma = \log \sum_{y' \in \mathcal{Y}} p_{\hat{\theta}(\mathcal{D}_N; x, y')}(y'|x). \tag{7}$$

*Proof.* The regret is equal for all choices of $y$. If we consider a different probability assignment, it should assign a smaller probability for at least one of the outcomes. If the true label is one of those outcomes it will lead to a higher regret. For more information see Fogel and Feder (2018). □

The pNML regret is associated with the model complexity (Zhang, 2012). This complexity measure formalizes the intuition that a model that fits almost every data pattern very well would be much more complex than a model that provides a relatively good fit to a small set of data. Thus, the pNML incorporates a trade-off between goodness of fit and model complexity as measured by the regret.

**Online update of a neural network.** Let $X_N$ and $Y_N$ be the data and label matrices of $N$ training points respectively

$$X_N = \begin{bmatrix} x_1 & x_2 & \dots & x_N \end{bmatrix}^\top \in \mathcal{R}^{N \times M}, \quad Y_N = \begin{bmatrix} y_1 & y_2 & \dots & y_N \end{bmatrix}^\top \in \mathcal{R}^{N \times C}, \tag{8}$$

such that the number of input features and model outputs are $M$ and $C$ respectively. Denote $X_N^+$ as the Moore–Penrose inverse of the data matrix

$$X_N^+ = \begin{cases} (X_N^\top X_N)^{-1} X_N^\top & Rank(X_N^\top X_N) = M \\ X_N^\top (X_N X_N^\top)^{-1} & otherwise, \end{cases} \tag{9}$$

$f(\cdot)$ and $f^{-1}(\cdot)$ as the activation and inverse activation functions, and $\theta \in \mathcal{R}^{M \times C}$ as the learnable parameters. The ERM, that minimizes the training set mean squared error (MSE), is

$$\hat{\theta}_N = X_N^+ f^{-1}(Y_N) = \arg\min_\theta ||Y_N - f(X_N \theta)||_2^2. \tag{10}$$

Recently, Zhuang et al. (2020) suggested a recursive formulation of the DNN weights. Using their scheme only one training sample is processed at a time: Denote the projection of a sample $x$ onto the orthogonal subspace of the training set correlation matrix as

$$x_\perp = \left(I - X_N^+ X_N\right) x, \tag{11}$$

the update rule when receiving a new training sample with data $x$ with label $y$ is

$$\hat{\theta}(\mathcal{D}_N; x, y) = \hat{\theta}_N + g\left(f^{-1}(y) - x^\top \hat{\theta}_N\right), \quad g \triangleq \begin{cases} \frac{1}{\|x_\perp\|^2} x_\perp & x_\perp \neq 0 \\ \frac{1}{1 + x^\top X_N^+ X_N^{+\top} x} X_N^+ X_N^{+\top} x, & x_\perp = 0 \end{cases}. \tag{12}$$

In their paper, Zhuang et al. (2020) applied this formulation for a DNN, layer by layer.

## 4 The pNML for a single layer NN

Intuitively, the pNML as stated in (7) can be described as follows: To assign a probability for a potential outcome, (i) add it to the training set with an arbitrary label, (ii) find the best-suited model, and (iii) take the probability it gives to the assumed label. Follow this procedure for every label and normalize to get a valid probability assignment. Use the log normalization factor as the confidence measure. This method can be extended to any general learning procedure that generates a prediction based on a training set. One such method is a single layer NN.

A single layer NN maps an input $x \in \mathcal{R}^{M \times 1}$ using the softmax function to a probability vector which represents the probability assignment to one of $C$ classes

$$p_\theta(i|x) = f(x^\top \theta)_i = \frac{e^{\theta_i^\top x}}{\sum_{j=1}^C e^{\theta_j^\top x}}, \quad i \in \{1, \ldots, C\}. \tag{13}$$

To align with the recursive formulation of (12), the label $y$ is a one-hot vector with $C$ elements and is represented by a row vector in the standard base $\{e_i\}_{i=1}^C$ (the i-th element is 1 and all others are 0). $c$ stands for the true test label such that $e_c$ is the one-hot vector of the true label. Also, the learnable parameters $\{\theta_i\}_{i=1}^C$ are the columns of the parameter matrix of (10).

We would like that the probability assignment of the true label to be 1. We define the following MSE minimization objective

$$\min_{\theta \in \Theta} \sum_{n=1}^N \left(1 - y_n f\left(\theta^\top x_n\right)\right)^2. \tag{14}$$

Minimizing this MSE objective is equivalent to constrain the genie to the following Gaussian family

$$P_\Theta = \left\{ p_\theta(y|x) = \frac{1}{\sqrt{2\pi\sigma^2}} \exp\left\{-\frac{1}{2\sigma^2} \left(e_c \left(y - f(x^\top \theta)\right)\right)^2\right\}, \theta \in R^{M \times C} \right\} \tag{15}$$

and minimizing the log loss with respect to it. More details are in appendix A. The inverse of the activation function is

$$z \triangleq f^{-1}\left(p_\theta\left(i|x\right)\right) = \ln p_\theta\left(i|x\right) + \ln \sum_{j=1}^C e^{\theta_j^\top x}. \tag{16}$$

To compute the genie, we assume that the label of the test sample is known and add it to the training set. We fit the model by optimizing the learnable parameters to minimize the loss of this new set.

**Lemma 1.** *Given test data $x$ with label $y = e_c$, the genie prediction of the true label is*

$$p_{\hat{\theta}(\mathcal{D}_N;x,y=e_c)}(c|x) = \frac{p_c}{p_c + p_c^{x^\top g}\left(1 - p_c\right)}, \tag{17}$$

*where $p_c$ is the probability assignment of the ERM model of the label $c$, and $g$ is as defined in (12).*

*Proof.* Using (12), the probability assignment of the genie can be written as follows.

$$p_{\hat{\theta}(\mathcal{D}_N;x,y=e_c)}(c|x) = \frac{e^{\hat{\theta}(\mathcal{D}_N;x,y=e_c)^\top x}}{\sum_{\substack{j=1 \\ j \neq c}}^C e^{\theta_j^\top x} + e^{\hat{\theta}(\mathcal{D}_N;x,y=e_c)^\top x}} = \frac{e^{x^\top \left[\theta_c + g\left(z - \theta_c^\top x\right)\right]}}{\sum_{j=1}^C e^{\theta_j^\top x} - e^{\theta_c^\top x} + e^{x^\top \left[\theta_c + g\left(z - \theta_c^\top x\right)\right]}}. \tag{18}$$

The genie knows the true test label $e_c$. The inverse activation function can be written as $z = \ln S$ where $S \triangleq \sum_{j=1}^C e^{\theta_j^\top x}$. The simplified numerator is

$$e^{\theta_c^\top x} e^{x^\top g\left(z - \theta_c^\top x\right)} = e^{\theta_c^\top x} \left[S e^{-\theta_c^\top x}\right]^{x^\top g} = S p_c^{-x^\top g} p_c. \tag{19}$$

Substituting to (18) and divide the numerator and denominator by $S$ provides the result. $\qquad \square$

The true test label is not available to a legit learner. Therefore, in the pNML process, every possible label is taken into account. The pNML regret is the logarithm of the sum of models' prediction, each one trained with a different test label value.

**Theorem 2.** *Denote $p_i$ as the ERM prediction of label $i$, the pNML regret of a single layer NN is*

$$\Gamma = \log \sum_{i=1}^{C} \frac{p_i}{p_i + p_i^{x^\top g}(1 - p_i)}. \tag{20}$$

*Proof.* The normalization factor is the sum of the probabilities assignment of models that were trained with a specific value of the test sample $K = \sum_{i=1}^{C} p_{\hat{\theta}(\mathcal{D}_N; x, y=e_i)}(i|x)$. Using Theorem 1, the log normalization factor is the pNML regret. With lemma 1, we get the explicit expression. $\qquad \square$

The pNML probability assignment of label $i \in \{1, \dots, C\}$ is the probability assignment of a model that was trained with that label divided by the normalization factor $q_{pNML}(i|x) = \frac{1}{K} p_{\hat{\theta}(\mathcal{D}_N; x, y)}(y|x)$.

Let $u_m$ and $h_m$ be the $m$-th eigenvector and eigenvalue of the training set data matrix $X_N$ such that for $x_\perp = 0$, the quantity $x^\top g$ is

$$x^\top g = \frac{x^\top X_N^+ X_N^{+\top} x}{1 + x^\top X_N^+ X_N^{+\top} x} = \frac{\frac{1}{N} \sum_{m=1}^{M} \frac{1}{h_m^2} \left(x^\top u_m\right)^2}{1 + \frac{1}{N} \sum_{i=1}^{M} \frac{1}{h_m^2} \left(x^\top u_m\right)^2}. \tag{21}$$

We make the following remarks.

1. If the test sample $x$ lies in the subspace spanned by the eigenvectors with large eigenvalues, $x^\top g$ is small and the corresponding regret is low $\lim_{x^\top g \to 0} \Gamma = \log \sum_{i=1}^{C} p_i = 0$. In this case, the pNML prediction is similar to the genie and can be trusted.

2. Test input that resides is in the subspace that corresponds to the small eigenvalues produces $x^\top g = 1$ and a large regret is obtained $\lim_{x^\top g \to 1} \Gamma = \log \sum_{i=1}^{C} \frac{1}{2 - p_i^2}$. The prediction for this test sample cannot be trusted. In section 5 we show that in this situation the test sample can be classified as an OOD sample.

3. As the training set size ($N$) increases $x^\top g$ becomes smaller and the regret decreases.

4. If the test sample is far from the decision boundary, the ERM assigns to one of the labels probability 1. In this case, the regret is 0 no matter in which subspace the test vector lies.

## 4.1 The pNML regret characteristics using a low-dimensional dataset

We demonstrate the characteristics of the derived regret and show in what situations the prediction of the test sample can be trusted. To visualize the pNML regret on a low-dimensional dataset, we use the Iris flower data set (Fisher, 1936). We utilize two classes and two features and name them $c_1$, $c_2$, and feature 1, feature 2 respectively.

Figure 1a shows the ERM probability assignment of class $c_2$ for a single layer NN that was fitted to the training data, which are marked in red. At the top left and bottom right, the model predicts with high probability that a sample from these areas belongs to class $c_1$ and $c_2$ respectively.

Figure 1b presents the analytical pNML regret. At the upper left and lower right, the regret is low: Although there are no training samples there, these regions are far from the decision boundary, adding one sample would not alter the probability assignment significantly, thus the pNML prediction is close to the genie. At the top right and bottom left, there are no training points therefore the regret is relatively high and the confidence in the prediction is low. In section 5, we show that these test samples, which are associated with high regret, can be classified as OOD samples.

In addition, we visualize the regret for overlapping classes. In figure 1c, the ERM probability assignment for inseparable class split is shown. The ERM probability is lower than 0.7 for all test feature values. Figure 1d presents the corresponding pNML regret. The pNML regret is small in the training data surroundings (including the mixed label area). The regret is large in areas where the training data is absent, as in the figure edges.

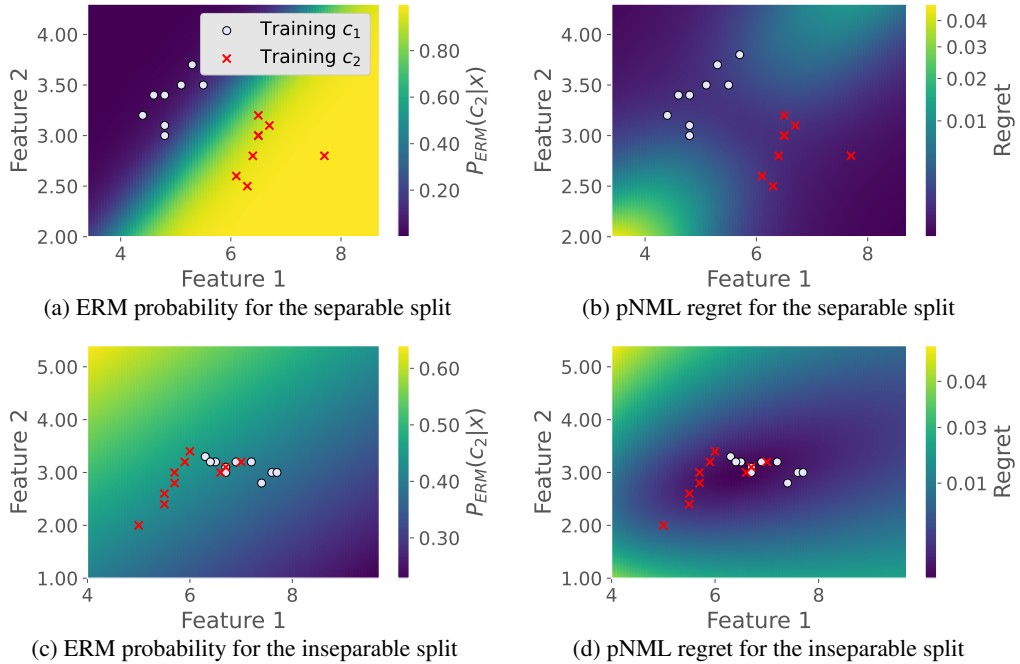

Figure 1: A single layer NN fitted to low dimensional data (the Iris flower set (Fisher, 1936)). (a) and (c) show the ERM probability assignment of class $c_2$ for a separable split and inseparable split respectively. The derived pNML regret for the separable split is shown in (b) and for the inseparable split is shown in (d). The training data of class $c_1$ and $c_2$ are marked with red circles and red crosses respectively. Low regret is associated with the training data surroundings. See section 4.1.

## 4.2 Deep neural network adaptation

In previous sections, we derived the pNML for a single layer NN. We next show that our derivations can, in fact, be applied to *any pretrained NN*, without requiring additional parameters or extra data.

First extract the embeddings of the training set: Denote $\phi(\cdot)$ as the embedding creation (feature extraction) using a pretrained ERM model, propagate the training samples through the DNN up to the last layer and compute the inverse of the data matrix $\phi(X_N)^+\phi(X_N)^{+\top}$. Then, given a specific test example $x$, extract its embedding $\phi(x)$ and its ERM probability assignment $\{p_i\}_{i=1}^C$. Finally calculate the regret as described in Theorem 1 using the training set and test embedding vectors.

We empirically found that norms of OOD embeddings are lower than those of IND samples. The regret depends on the norm of the test sample: For $0 < a < b$, the regret of $ax$ is lower than the regret of $bx$. Hence, we normalize all embeddings (training, IND, and OOD) to have $L^2$ norms equal to 1.0.

Samples with a high regret value are considered samples with a large distance from the genie, the learner that knows the true label, and therefore the prediction cannot be trusted. Our proposed method utilizes this regret value to determine whether a test data item represents a known or an unknown.

## 5 Application to out-of-distribution detection

We rigorously test the effectiveness of the pNML regret for OOD detection[1]. The motivation for using the individual setting and the pNML as its solution for OOD detection is that in the individual setting there is no assumption on the way the data is generated. The absence of assumption means that the result holds for a wide range of scenarios (PAC, stochastic, and even adversary) and specifically to OOD detection, where the OOD samples are drawn from an unknown distribution.

---

[1]Code is available in `https://github.com/kobybibas/pnml_ood_detection`

Table 1: OOD detection for DenseNet-BC-100 model, comparing our pNML-based approach to leading methods. Results reported using AUROC show our method enhances previous work up to 69.4%, 49.4%, 3.1%, and 2.2%. See section 5.2 for more details.

| IND | OOD | Baseline/+pNML | ODIN/+pNML | Gram/+pNML | OECC/+pNML |
|---|---|---|---|---|---|
| CIFAR-100 | iSUN | 69.7 / **96.4** | 84.5 / **96.7** | 99.0 / **99.5** | 99.2 / **99.5** |
| | LSUN (R) | 70.8 / **96.6** | 86.0 / **96.9** | 99.3 / **99.7** | 99.4 / **99.6** |
| | LSUN (C) | 80.1 / **93.1** | 91.5 / **93.1** | 91.4 / **94.5** | 93.9 / **96.1** |
| | Imagenet (R) | 71.6 / **97.4** | 85.5 / **97.6** | 99.0 / **99.5** | 99.0 / **99.5** |
| | Imagenet (C) | 76.2 / **95.7** | 88.8 / **96.0** | 97.7 / **98.7** | 98.2 / **99.0** |
| | Uniform | 43.3 / **100** | 83.7 / **100** | 100 / **100** | 99.9 / **100** |
| | Gaussian | 30.6 / **100** | 50.6 / **100** | 100 / **100** | 100 / **100** |
| | SVHN | 82.6 / **96.2** | 92.5 / **96.2** | 97.3 / **98.4** | 97.0 / **97.5** |
| CIFAR-10 | iSUN | 94.8 / **98.7** | 98.9 / **98.9** | 99.8 / **100** | 99.9 / **100** |
| | LSUN (R) | 95.5 / **98.9** | 99.2 / **99.2** | 99.9 / **100** | 99.9 / **100** |
| | LSUN (C) | 93.0 / **96.4** | 95.8 / **96.4** | 97.5 / **98.7** | 98.9 / **99.9** |
| | Imagenet (R) | 94.1 / **98.8** | 98.5 / **99.0** | 99.7 / **99.9** | 99.8 / **99.9** |
| | Imagenet (C) | 93.8 / **97.7** | 97.6 / **97.9** | 99.3 / **99.7** | 99.5 / **99.9** |
| | Uniform | 96.6 / **100** | 100 / **100** | 100 / **100** | 100 / **100** |
| | Gaussian | 97.6 / **100** | 100 / **100** | 100 / **100** | 100 / **100** |
| | SVHN | 89.9 / **98.4** | 94.6 / **98.7** | 99.1 / **99.6** | 99.6 / **100** |
| SVHN | iSUN | 94.4 / **98.7** | 92.8 / **99.1** | 99.8 / **99.9** | 100 / **100** |
| | LSUN (R) | 94.1 / **98.4** | 92.5 / **98.9** | 99.8 / **100** | 100 / **100** |
| | LSUN (C) | 92.9 / **98.0** | 88.6 / **98.1** | 98.6 / **99.4** | 99.8 / **100** |
| | Imagenet (R) | 94.8 / **98.6** | 93.3 / **99.0** | 99.7 / **99.9** | 100 / **100** |
| | Imagenet (C) | 94.6 / **98.6** | 92.8 / **98.8** | 99.4 / **99.8** | 100 / **100** |
| | Uniform | 93.2 / **99.8** | 91.6 / **100** | 99.9 / **100** | 100 / **100** |
| | Gaussian | 97.4 / **99.8** | 98.9 / **99.9** | 100 / **100** | 100 / **100** |
| | CIFAR-10 | 91.8 / **96.7** | 88.9 / **97.8** | 95.4 / **97.3** | 99.5 / **100** |
| | CIFAR-100 | 91.4 / **96.7** | 88.2 / **97.8** | 96.4 / **98.0** | 99.6 / **100** |

## 5.1 Experimental setup

We follow the standard experimental setup (Liu et al., 2020; Sastry and Oore, 2020; Lee et al., 2018). All the assets we used are open-sourced with either Apache-2.0 License or Attribution-NonCommercial 4.0 International licenses. We ran all experiments on NVIDIA K80 GPU.

**IND sets.** For datasets that represent known classes, we use CIFAR-100, CIFAR-10 (Krizhevsky et al., 2014) and SVHN (Netzer et al., 2011). These sets contain RGB images with 32x32 pixels. In addition, to evaluate higher resolution images, we use ImageNet-30 set (Hendrycks et al., 2019b).

**OOD sets.** The OOD sets are represented by TinyImageNet (Liang et al., 2018), LSUN (Yu et al., 2015), iSUN (Xu et al., 2015), Uniform noise images, and Gaussian noise images. We use two variants of TinyImageNet and LSUN sets: a 32x32 image crop that is represented by "(C)" and a resizing of the images to 32x32 pixels that termed by "(R)". We also used CIFAR-100, CIFAR-10, and SVHN as OOD for models that were not trained with them.

**Evaluation methodology.** We benchmark our approach by adopting the following metrics (Sastry and Oore, 2020; Lee et al., 2018): (i) AUROC: The area under the receiver operating characteristic curve of a threshold-based detector. A perfect detector corresponds to an AUROC score of 100%. (ii) TNR at 95% TPR: The probability that an OOD sample is correctly identified (classified as negative) when the true positive rate equals 95%. (iii) Detection accuracy: Measures the maximum possible classification accuracy over all possible thresholds.

## 5.2 Results

We build upon the existing leading methods: Baseline (Hendrycks and Gimpel, 2017), ODIN (Liang et al., 2018), Gram (Sastry and Oore, 2020), OECC (Papadopoulos et al., 2021), and Energy (Liu et al., 2020). We use the following pretrained models: ResNet-34 (He et al., 2016), DenseNet-BC-

Table 2: A comparison of our pNML regret based detection to leading methods for ResNet-34 model. Results measured by the AUROC metric show that our technique offers significant improvements over previous work of up to 41.5%, 11.9%, 2.4%, and 2.1%. See section 5.2 for more details.

| IND | OOD | Baseline/+pNML | ODIN/+pNML | Gram/+pNML | OECC/+pNML |
|---|---|---|---|---|---|
| CIFAR-100 | iSUN | 75.7 / **83.0** | 85.6 / **87.6** | 98.8 / **99.1** | 99.0 / **99.3** |
| | LSUN (R) | 75.6 / **83.8** | 85.4 / **88.0** | 99.2 / **99.4** | 99.3 / **99.6** |
| | LSUN (C) | 75.5 / **83.1** | 82.6 / **88.1** | 92.2 / **94.6** | 95.7 / **97.8** |
| | Imagenet (R) | 77.1 / **84.4** | 87.7 / **88.5** | 98.9 / **99.2** | 98.7 / **98.9** |
| | Imagenet (C) | 79.6 / **85.8** | 85.6 / **88.6** | 97.7 / **98.4** | 97.9 / **98.1** |
| | Uniform | 85.2 / **98.1** | 99.0 / **99.4** | 100 / **100** | 100 / **100** |
| | Gaussian | 45.0 / **86.5** | 83.8 / **95.7** | 100 / **100** | 100 / **100** |
| | SVHN | 79.3 / **90.9** | 94.0 / **95.4** | 96.0 / **97.9** | 97.0 / **97.6** |
| CIFAR-10 | iSUN | 91.0 / **96.4** | 94.0 / **97.5** | 99.8 / **100** | 99.9 / **99.9** |
| | LSUN (R) | 91.1 / **96.6** | 94.1 / **97.7** | 99.9 / **100** | **100** / 99.9 |
| | LSUN (C) | 91.8 / **95.4** | 93.6 / **95.6** | 97.9 / **99.1** | 99.1 / **99.5** |
| | Imagenet (R) | 91.0 / **95.4** | 93.9 / **96.6** | 99.7 / **99.9** | 99.9 / **99.9** |
| | Imagenet (C) | 91.4 / **95.4** | 93.3 / **96.2** | 99.3 / **99.7** | 99.7 / **99.8** |
| | Uniform | 96.1 / **99.8** | 99.9 / **100** | 100 / **100** | 100 / **100** |
| | Gaussian | 97.5 / **100** | 100 / **100** | 100 / **100** | 100 / **100** |
| | SVHN | 89.9 / **95.1** | 95.8 / **97.9** | 99.5 / **99.8** | 99.8 / **99.8** |
| SVHN | iSUN | 92.2 / **97.1** | 91.4 / **98.0** | 99.8 / **99.9** | 100 / **100** |
| | LSUN (R) | 91.5 / **96.7** | 90.6 / **97.7** | 99.8 / **100** | 100 / **100** |
| | LSUN (C) | 92.8 / **97.0** | 92.3 / **97.1** | 98.8 / **99.6** | 99.7 / **99.9** |
| | Imagenet (R) | 93.5 / **97.5** | 92.8 / **98.3** | 99.8 / **99.9** | 100 / **100** |
| | Imagenet (C) | 94.2 / **97.5** | 93.7 / **98.2** | 99.5 / **99.9** | 99.9 / **100** |
| | Uniform | 96.0 / **98.5** | 95.5 / **99.5** | 100 / **100** | 100 / **100** |
| | Gaussian | 96.1 / **98.4** | 96.1 / **99.6** | 100 / **100** | 100 / **100** |
| | CIFAR-10 | 93.0 / **97.4** | 92.0 / **98.0** | 97.4 / **99.3** | 99.4 / **99.8** |
| | CIFAR-100 | 92.5 / **97.1** | 91.7 / **97.8** | 97.5 / **99.2** | 99.4 / **99.8** |

100 (Huang et al., 2017) and WideResNet-40 (Zagoruyko and Komodakis, 2016). Training was performed using CIFAR-100, CIFAR-10 and SVHN, each training set used separately to provide a complete picture of our proposed method's capabilities. Notice that ODIN, OECC, and Energy methods use OOD sets during training and the Gram method requires IND validation samples.

Table 1 and Table 2 show the AUROC of different OOD sets for DenseNet and ResNet models respectively. Our approach improves all the compared methods in nearly all combinations of IND-OOD sets. The largest AUROC gain over the current state-of-the-art is of CIFAR-100 as IND and LSUN (C) as OOD: For the DenseNet model, we improve Gram and OECC method by 3.1% and 2.2% respectively. For the ResNet model, we improve this combination by 2.4% and 2.1% respectively. The additional metrics (TNR at 95% FPR and detection accuracy) are shown in appendix E.

The Baseline method uses a pretrained ERM model with no extra data. Combining the pNML regret with the standard ERM model as shown in the Baseline+pNML column surpasses Baseline by up to 69.4% and 41.5% for DensNet and ResNet, respectively. Also, Baseline+pNML is comparable to the more sophisticated methods: Although it lacks tunable parameters and does not use extra data, Baseline+pNML outperforms ODIN in most DenseNet IND-OOD set combinations.

Table 3 shows the results of the Energy method and our method when combined with the pretrained Energy model (Energy+pNML) with WideResNet-40 model on CIFAR-100 and CIFAR-10 as IND sets. Evidently, our method improves the AUROC of the OOD detection task in 14 out of 16 IND-OOD combinations. The most significant improvement is in CIFAR-100 as IND and ImageNet (R) and iSUN as the OOD sets. In these sets, we improve the AUROC by 15.6% and 15.2% respectively. For TNR at TPR 95%, the pNML regret enhances the CIFAR-100 and Gaussian combination by 90.4% and achieves a perfect separation of IND-OOD samples.

For high resolution images, we use Resnet-18 and ResNet-101 models trained on ImageNet. We utilize the ImageNet-30 training set for computing $\phi(X_N)^+\phi(X_N)^{+\top}$. All images were resized to $254 \times 254$ pixels. We compare the result to the Baseline method in Table 4. The table shows that the pNML outperforms Baseline by up to 9.8% and 8.23% for ResNet-18 and ResNet-101 respectively.

Table 3: OOD detection for WideResNet-40 model. Results show our method improves the Energy (Liu et al., 2020) method up to 15.6%, 90.4%, and 15.9% for AUROC, TNR at TPR 95%, and Detection accuracy respectively. See section 5.2 for more details.

| IND | OOD | AUROC | TNR at TPR 95% | Detection Acc. |
|---|---|---|---|---|
| | | | Energy/+pNML | |
| CIFAR-100 | iSUN | 78.4 / **93.6** | 30.7 / **62.5** | 71.1 / **87.0** |
| | LSUN (R) | 80.3 / **94.1** | 31.2 / **65.5** | 73.1 / **87.5** |
| | LSUN (C) | **95.9** / 95.5 | **80.0** / 79.3 | **89.3** / 89.1 |
| | Imagenet (R) | 71.4 / **87.0** | 22.1 / **44.8** | 66.1 / **79.9** |
| | Imagenet (C) | 79.7 / **87.3** | 36.9 / **49.4** | 72.8 / **79.7** |
| | Uniform | 97.9 / **99.8** | 95.2 / **100** | 95.8 / **99.6** |
| | Gaussian | 92.0 / **99.8** | 9.6 / **100** | 92.3 / **99.8** |
| | SVHN | **96.5** / 96.4 | 79.2 / **82.8** | 90.5 / **91.3** |
| CIFAR-10 | iSUN | 99.3 / **99.4** | 98.3 / **98.7** | 96.7 / **97.0** |
| | LSUN (R) | 99.3 / **99.5** | 98.6 / **99.0** | 97.0 / **97.3** |
| | LSUN (C) | 99.4 / **99.5** | 98.6 / **98.6** | 97.0 / **97.1** |
| | Imagenet (R) | 98.1 / **98.1** | 92.0 / **92.4** | 94.0 / **94.0** |
| | Imagenet (C) | 98.6 / **98.6** | 94.4 / **94.6** | 94.9 / **94.9** |
| | Uniform | 99.0 / **99.9** | 100 / **100** | 98.7 / **99.8** |
| | Gaussian | 99.1 / **99.9** | 100 / **100** | 98.7 / **99.8** |
| | SVHN | 99.3 / **99.6** | 98.3 / **98.9** | 96.9 / **97.6** |

Table 4: OOD detection with ImageNet-30 as the IND set. Results show that the pNML outperforms the Baseline (Hendrycks and Gimpel, 2017) method up to 9.8% and 8.23% for ResNet-18 and ResNet-101 models respectively. See section 5.2 for more details.

| IND | OOD | ResNet-18 | ReseNet-101 |
|---|---|---|---|
| | | Baseline/+pNML | Baseline/+pNML |
| ImageNet-30 | iSUN | 95.58 / **99.74** | 96.26 / **99.54** |
| | LSUN (R) | 95.51 / **99.72** | 95.77 / **99.43** |
| | LSUN (C) | 96.89 / **99.77** | 98.00 / **99.86** |
| | Uniform | 99.35 / **99.99** | 98.70 / **100** |
| | Gaussian | 98.78 / **100** | 98.61 / **100** |
| | SVHN | 99.18 / **99.99** | 98.94 / **99.98** |
| | CIFAR-10 | 89.99 / **99.79** | 91.24 / **99.47** |
| | CIFAR-100 | 92.15 / **92.15** | 93.39 / **99.58** |

# 6 Conclusions

We derived the analytical expression of the pNML regret for a single layer NN. We showed that the model generalizes well when the test data resides in either a subspace spanned by the eigenvectors associated with the large eigenvalues of the training data correlation matrix or far from the decision boundary. We showed how to apply the pNML for any pretrained DNN that uses the softmax layer with neither additional parameters nor extra data. We demonstrated the effectiveness of our pNML regret–based approach on 74 IND-OOD detection benchmarks. Compared with the recent state of the art methods, our approach elevates absolute AUROC values by up to 15.6%

For future work, in our regret derivation we constrained the genie hypothesis set to the Gaussian family and a single layer NN. We would like to extend the derivation to larger hypothesis sets and more layers, believing it would improve performance. Furthermore, the pNML regret can be used for additional tasks such as active learning, probability calibration, and adversarial attack detection.

**Societal impacts.** The negative social impact of this work depends on the application. For instance, in a surveillance camera scenario, a person from a minority group can be flagged as OOD if the minority group was not included in the training set. We recommend that when OOD is flagged a human will intervene rather than an algorithmic response since the nature of the OOD is unknown.

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
