# Single Layer Predictive Normalized Maximum Likelihood for Out-of-Distribution Detection –Supplementary material–

## A  MSE minimization is equivalent to log-loss minimization

We use the same notations as in section 4.

Denote $e_c$ as a one-hot row vector of the true label, we define the hypothesis set that genie is allowed to choose from as

$$P_\Theta = \left\{ p_\theta(y|x) = \frac{1}{\sqrt{2\pi\sigma^2}} \exp\left\{ -\frac{1}{2\sigma^2} \left[ \left(y - f(x_n^\top \theta)\right) e_c^\top \right]^2 \right\} \right\}. \tag{1}$$

The genie chooses the learner from the hypothesis set that minimizes the log-loss. Let $x_n \in \mathcal{R}^{M \times 1}$ be the $n$-th data with the label $c_n \in \{1, 2, \ldots, C\}$, $y_n$ be a row vector where $y_{nc_n}$ is its $c_n$ element. We show that the log-loss minimizer of this hypothesis set is equal to the MSE minimizer:

$$\begin{aligned}
\underset{\theta \in \mathcal{R}^{M \times C}}{\arg\min} \ell(p_\theta, X_N, Y_N) &= \underset{\theta \in \mathcal{R}^{M \times C}}{\arg\min} \left[ -\log \prod_{n=1}^{N} \frac{1}{\sqrt{2\pi\sigma^2}} \exp\left\{ -\frac{1}{2\sigma^2} \left(y_{nc_n} - f(x_n^\top \theta)_{c_n}\right)^2 \right\} \right] \\
&= \underset{\theta \in \mathcal{R}^{M \times C}}{\arg\min} \sum_{n=1}^{N} \left(y_{nc_n} - f(x_n^\top \theta)_{c_n}\right)^2.
\end{aligned} \tag{2}$$

We know that the training set label are one-hot vector $y_n = e_{c_n}$ such that $y_{nc_n} = 1$:

$$\begin{aligned}
\underset{\theta \in \mathcal{R}^{M \times C}}{\arg\min} \ell(p_\theta, X_N, Y_N) &= \underset{\theta \in \mathcal{R}^{M \times C}}{\arg\min} \sum_{n=1}^{N} \left(1 - f(x_n^\top \theta)_{c_n}\right)^2 = \underset{\theta \in \mathcal{R}^{M \times C}}{\arg\min} \sum_{n=1}^{N} \left\| 1 - f(x_n^\top \theta) e_{c_n}^\top \right\|_2^2 \\
&= \underset{\theta \in \mathcal{R}^{M \times C}}{\arg\min} \sum_{n=1}^{N} \left\| 1 - y_n f(x_n^\top \theta) \right\|_2^2
\end{aligned} \tag{3}$$

which is the MSE minimization objective we defined in section 4.

## B  Single layer NN pNML regret simulation

We simulate the response of the pNML regret for two classes (C=2) and divide it by $\log C$ to have the regret bounded between 0 and 1. Figure 1 shows the regret behaviour for different $p_1$ (the ERM probability assignment of class 1) as a function of $x^\top g$.

For an ERM model that is certain on the prediction ($p_1 = 0.99$ that is represented by the purple curve), a slight variation of $x^\top g$ causes a large response of the regret comparing to $p_1$ that equals 0.55 and 0.85. All curves converging to the maximal regret for $x^\top g$ greater than 6.

Submitted to 35th Conference on Neural Information Processing Systems (NeurIPS 2021). Do not distribute.

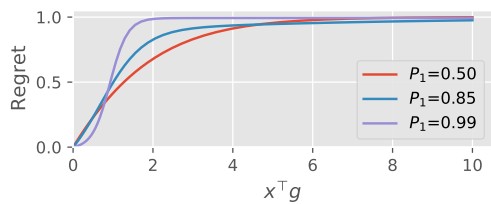

Figure 1: The pNML regret for a two class predictor. $p_1$ is the ERM prediction of class $c_1$.

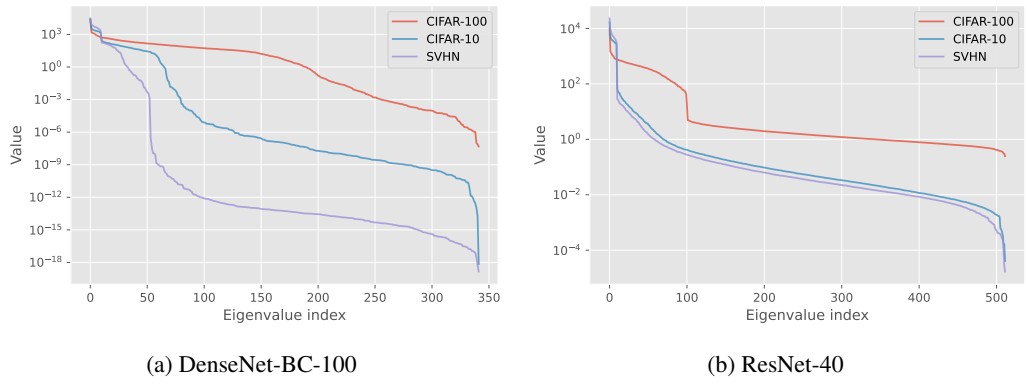

(a) DenseNet-BC-100

(b) ResNet-40

Figure 2: The spectrum of the training embeddings.

## C    The spectrum of real dataset

We provide a visualization of the training data spectrum when propagated to the last layer of a DNN.

We feed the training data through the model up to the last layer to create the training embeddings. Next, we compute the correlation matrix of the training embeddings and perform an SVD decomposition. We plot the eigenvalues for different training sets in figure 2.

Figure 2a shows the eigenvalues of DenseNet-BC-100 model when ordered from the largest to smallest. For the SVHN training set, most of the energy is located in the first 50 eigenvalues and then there is a significant decrease of approximately $10^3$. The same phenomenon is also seen in figure 2a that shows the eigenvalues of ResNet-40 model. In our derived regret, if the test sample is located in the subspace that is associated with small eigenvalues (for example indices 50 or above for DenseNet trained with SVHN) then $x^\top g$ is large and so is the pNML regret.

For both DensNet and ResNet models, the values of the eigenvalues of CIFAR-100 seem to be spread more evenly compared to CIFAR-10, and the CIFAR-10 are more uniform than the SVHN. How much the eigenvalues are spread can indicate the variability of the set: SVHN is a set of digits that is much more constrained than CIFAR-100 which has 100 different classes.

## D    Gram vs. Gram+pNML

We further explore the benefit of the pNML regret in detecting OOD samples over the Gram approach. We focus on the DenseNet model with CIFAR-100 as the training set and LSUN (C) as the OOD set.

Figure 3a shows the 2D histogram of the IND set based on the pNML regret values and Gram scores. In addition, we plotted the best threshold for separating the IND and OOD of these sets. pNML regret values less than 0.0024 and Gram scores below 0.0017 qualify as IND samples by both the pNML and Gram scores. Gram and Gram+pNML do not succeed to classify 1205 and 891 out of a total 10,000 IND samples respectively.

Figure 3b presents the 2D histogram of the LSUN (C) as OOD set. For regret values greater than 0.0024 and Gram score lower than 0.0017, the pNML succeeds to classify as IND but the Gram fails:

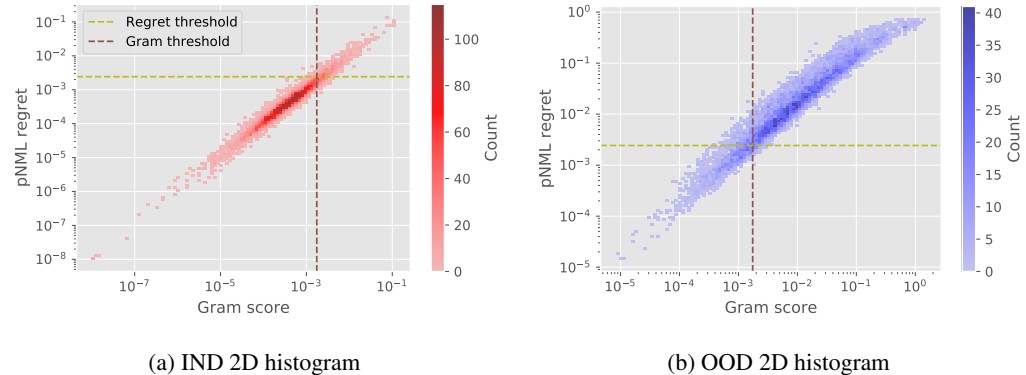

(a) IND 2D histogram

(b) OOD 2D histogram

Figure 3: 2D histogram of the pNML regret and the Gram score of a DenseNet model trained with CIFAR-100 as IND set and LSUN (C) as OOD.

Table 1: DenseNet-BC-100 model TNR at TPR95% comparison. The compared methods are Baseline (Hendrycks and Gimpel, 2017), ODIN (Liang et al., 2018), Gram (Sastry and Oore, 2020), and OECC (Papadopoulos et al., 2021)

| IND | OOD | Baseline/+pNML | ODIN/+pNML | Gram/+pNML | OECC/+pNML |
|---|---|---|---|---|---|
| CIFAR-100 | iSUN | 14.8 / **81.2** | 37.4 / **82.8** | 95.8 / **97.9** | 97.5 / **99.2** |
| | LSUN (R) | 16.4 / **82.7** | 41.6 / **84.5** | 97.1 / **98.7** | 98.4 / **99.6** |
| | LSUN (C) | 28.3 / 65.7 | 58.2 / 65.4 | 65.3 / **76.3** | 74.6 / **83.4** |
| | Imagenet (R) | 17.3 / **86.4** | 43.0 / **87.9** | 95.6 / **98.0** | 96.5 / **99.0** |
| | Imagenet (C) | 24.3 / **77.2** | 52.5 / **78.6** | 88.8 / **93.8** | 92.6 / **96.9** |
| | Uniform | 0.0 / **100** | 0.0 / **100** | 100 / **100** | 100 / **100** |
| | Gaussian | 0.0 / **100** | 0.0 / **100** | 100 / **100** | 100 / **100** |
| | SVHN | 26.2 / **79.2** | 56.8 / **79.0** | 89.3 / **93.7** | 89.0 / **90.7** |
| CIFAR-10 | iSUN | 63.3 / **93.2** | 94.0 / **94.3** | 99.1 / **99.8** | 99.7 / **100** |
| | LSUN (R) | 66.9 / **94.2** | **96.2** / 95.8 | 99.5 / **99.9** | 99.8 / **100** |
| | LSUN (C) | 52.0 / **79.9** | 74.6 / **80.2** | 88.7 / **94.4** | 95.7 / **99.6** |
| | Imagenet (R) | 59.4 / **93.4** | 92.5 / **94.6** | 98.8 / **99.6** | 99.3 / **99.9** |
| | Imagenet (C) | 57.0 / **87.1** | 86.9 / **88.3** | 96.8 / **98.7** | 98.6 / **99.8** |
| | Uniform | 76.4 / **100** | 100 / **100** | 100 / **100** | 100 / **100** |
| | Gaussian | 88.1 / **100** | 100 / **100** | 100 / **100** | 100 / **100** |
| | SVHN | 40.4 / **92.2** | 77.0 / **95.0** | 96.0 / **98.2** | 98.5 / **99.9** |
| SVHN | iSUN | 78.3 / **93.6** | 78.5 / **96.3** | 99.6 / **99.9** | 100 / **100** |
| | LSUN (R) | 77.1 / **91.7** | 77.0 / **95.2** | 99.7 / **100** | 100 / **100** |
| | LSUN (C) | 73.5 / **89.7** | 68.5 / **90.0** | 93.4 / **97.2** | 99.5 / **100** |
| | Imagenet (R) | 79.7 / **93.6** | 79.0 / **95.8** | 99.2 / **99.8** | 100 / **100** |
| | Imagenet (C) | 78.9 / **92.8** | 77.6 / **94.5** | 98.0 / **99.3** | 99.9 / **100** |
| | Uniform | 66.1 / **100** | 71.7 / **100** | 100 / **100** | 100 / **100** |
| | Gaussian | 88.7 / **99.7** | 95.6 / **100** | 100 / **100** | 100 / **100** |
| | CIFAR-10 | 69.1 / **81.0** | 66.6 / **88.5** | 75.1 / **86.8** | 98.9 / **100** |
| | CIFAR-100 | 68.7 / **81.4** | 65.7 / **88.5** | 80.3 / **90.1** | 99.1 / **100** |

There are 473 samples that the pNML classifies as OOD but the Gram fails, in contrast to 76 samples classified as such by the Gram and not by the pNML regret. Most of the pNML improvement is in assigning a high score to OOD samples while there is not much change in the rank of the IND ones.

# E Additional out of distribution metrics

The additional OOD metrics, TNR at 95% FPR and Detection Accuracy, for the DensNet model are shown in table 1 and table 2 respectively and for the ResNet are presented in table 3 and table 4. We improve the compared methods for all IND-OOD sets except for 6 experiments of ODIN method with the TNR at 95% metric. We show the TNR vs FPR of these experiments in figure 4. We state that for most of the TNR values, the pNML regret outperforms the ODIN method, as also shown in the AUROC metric.

Table 2: DenseNet-BC-100 model Detection Acc. comparison. The compared methods are Baseline (Hendrycks and Gimpel, 2017), ODIN (Liang et al., 2018), Gram (Sastry and Oore, 2020), and OECC (Papadopoulos et al., 2021)

| IND | OOD | Baseline/+pNML | ODIN/+pNML | Gram/+pNML | OECC/+pNML |
|---|---|---|---|---|---|
| CIFAR-100 | iSUN | 64.0 / **89.9** | 76.5 / **90.3** | 95.6 / **97.0** | 96.5 / **98.0** |
| | LSUN (R) | 65.0 / **90.5** | 77.7 / **91.0** | 96.3 / **97.4** | 97.2 / **98.5** |
| | LSUN (C) | 72.6 / **85.3** | **83.4** / 85.2 | 83.7 / **87.5** | 87.0 / **90.2** |
| | Imagenet (R) | 65.7 / **91.6** | 77.3 / **92.1** | 95.5 / **97.0** | 96.0 / **97.8** |
| | Imagenet (C) | 69.0 / **89.0** | 80.8 / **89.3** | 92.4 / **94.5** | 94.0 / **96.1** |
| | Uniform | 64.2 / **100** | 85.0 / **100** | 100 / **100** | 99.9 / **100** |
| | Gaussian | 58.8 / **100** | 66.9 / **100** | 100 / **100** | 100 / **100** |
| | SVHN | 75.5 / **90.3** | 86.0 / **90.3** | 92.3 / **94.4** | 92.1 / **93.0** |
| CIFAR-10 | iSUN | 89.2 / **94.2** | 94.6 / **94.8** | 98.0 / **99.0** | 98.7 / **99.6** |
| | LSUN (R) | 90.2 / **94.7** | **95.6** / 95.5 | 98.6 / **99.3** | 98.9 / **99.7** |
| | LSUN (C) | 86.9 / **89.5** | **89.7** / 89.4 | 92.1 / **94.8** | 95.5 / **98.8** |
| | Imagenet (R) | 88.5 / **94.3** | 94.0 / **94.9** | 97.9 / **98.8** | 98.3 / **99.2** |
| | Imagenet (C) | 88.0 / **91.9** | **92.3** / 92.2 | 96.2 / **97.7** | 97.4 / **99.0** |
| | Uniform | 94.8 / **100** | 99.7 / **100** | 100 / **100** | 100 / **100** |
| | Gaussian | 95.3 / **100** | 99.8 / **100** | 100 / **100** | 100 / **100** |
| | SVHN | 83.2 / **94.0** | 88.1 / **95.1** | 95.8 / **97.3** | 97.4 / **99.3** |
| SVHN | iSUN | 89.7 / **94.6** | 87.7 / **95.7** | 98.3 / **99.1** | 99.8 / **100** |
| | LSUN (R) | 89.2 / **93.8** | 87.2 / **95.1** | 98.6 / **99.2** | 99.9 / **100** |
| | LSUN (C) | 88.0 / **92.8** | 83.6 / **92.8** | 94.3 / **96.4** | 98.5 / **99.8** |
| | Imagenet (R) | 90.2 / **94.4** | 88.2 / **95.5** | 97.9 / **98.9** | 99.7 / **100** |
| | Imagenet (C) | 89.8 / **94.2** | 87.6 / **94.8** | 96.7 / **98.1** | 99.5 / **100** |
| | Uniform | 87.9 / **98.8** | 85.2 / **99.4** | 99.9 / **100** | 100 / **100** |
| | Gaussian | 93.6 / **98.4** | 95.4 / **99.1** | 100 / **100** | 100 / **100** |
| | CIFAR-10 | 86.5 / **91.0** | 83.5 / **92.7** | 89.0 / **92.0** | 97.4 / **99.8** |
| | CIFAR-100 | 86.5 / **91.0** | 83.1 / **92.8** | 90.4 / **93.2** | 97.7 / **99.8** |

Table 3: ResNet-34 model TNR at TPR95% comparison. The compared methods are Baseline (Hendrycks and Gimpel, 2017), ODIN (Liang et al., 2018), Gram (Sastry and Oore, 2020), and OECC (Papadopoulos et al., 2021)

| IND | OOD | Baseline/+pNML | ODIN/+pNML | Gram/+pNML | OECC/+pNML |
|---|---|---|---|---|---|
| CIFAR-100 | iSUN | 16.6 / **26.1** | **45.4** / 44.1 | 94.7 / **95.7** | 97.2 / **98.0** |
| | LSUN (R) | 18.4 / **28.4** | **45.5** / 44.6 | 96.6 / **97.1** | 98.3 / **99.0** |
| | LSUN (C) | 18.2 / **30.1** | 44.0 / **51.2** | 64.6 / **72.9** | 80.3 / **89.8** |
| | Imagenet (R) | 20.2 / **31.8** | **48.7** / 47.6 | 94.8 / **96.2** | 95.5 / **95.8** |
| | Imagenet (C) | 23.9 / **33.6** | 44.4 / **48.1** | 88.3 / **91.6** | 90.6 / **91.6** |
| | Uniform | 10.1 / **89.1** | 98.4 / **98.5** | 100 / **100** | 100 / **100** |
| | Gaussian | 0.0 / **13.7** | 4.5 / **66.8** | 100 / **100** | 100 / **100** |
| | SVHN | 19.9 / **52.0** | 63.8 / **75.0** | 80.3 / **89.0** | 86.8 / **89.2** |
| CIFAR-10 | iSUN | 44.5 / **78.5** | 73.0 / **86.3** | 99.4 / **99.9** | 99.8 / **100** |
| | LSUN (R) | 45.1 / **79.8** | 73.5 / **87.5** | 99.6 / **99.9** | 99.9 / **100** |
| | LSUN (C) | 48.0 / **72.6** | 63.1 / **76.1** | 90.2 / **95.9** | 96.3 / **98.9** |
| | Imagenet (R) | 44.0 / **72.8** | 71.8 / **81.9** | 98.9 / **99.6** | 99.6 / **99.8** |
| | Imagenet (C) | 45.9 / **71.4** | 66.5 / **78.0** | 97.0 / **98.8** | 98.9 / **99.7** |
| | Uniform | 71.4 / **100** | 100 / **100** | 100 / **100** | 100 / **100** |
| | Gaussian | 90.2 / **100** | 100 / **100** | 100 / **100** | 100 / **100** |
| | SVHN | 32.2 / **69.1** | 81.9 / **90.8** | 97.6 / **99.2** | 99.3 / **99.7** |
| SVHN | iSUN | 77.0 / **85.6** | 79.1 / **90.6** | 99.5 / **99.9** | 100 / **100** |
| | LSUN (R) | 74.4 / **82.9** | 76.6 / **88.3** | 99.6 / **99.9** | 100 / **100** |
| | LSUN (C) | 76.1 / **86.3** | 78.5 / **86.4** | 94.5 / **98.4** | 99.3 / **99.9** |
| | Imagenet (R) | 79.0 / **88.0** | 80.8 / **92.5** | 99.4 / **99.8** | 100 / **100** |
| | Imagenet (C) | 80.4 / **88.4** | 82.4 / **91.5** | 98.6 / **99.7** | 99.9 / **100** |
| | Uniform | 85.2 / **95.6** | 86.1 / **99.3** | 100 / **100** | 100 / **100** |
| | Gaussian | 84.8 / **94.9** | 90.9 / **99.4** | 100 / **100** | 100 / **100** |
| | CIFAR-10 | 78.3 / **87.2** | 79.9 / **90.4** | 86.1 / **97.2** | 98.4 / **99.8** |
| | CIFAR-100 | 76.9 / **85.8** | 78.5 / **89.1** | 87.6 / **96.9** | 98.4 / **99.8** |

Table 4: ResNet-34 model Detection Acc. comparison. The compared methods are Baseline (Hendrycks and Gimpel, 2017), ODIN (Liang et al., 2018), Gram (Sastry and Oore, 2020), and OECC (Papadopoulos et al., 2021)

| IND | OOD | Baseline/+pNML | ODIN/+pNML | Gram/+pNML | OECC/+pNML |
|---|---|---|---|---|---|
| CIFAR-100 | iSUN | 70.1 / **76.0** | 78.6 / **79.3** | 95.0 / **95.4** | 96.2 / **96.9** |
| | LSUN (R) | 69.8 / **76.5** | 78.1 / **79.8** | 96.0 / **96.2** | 96.9 / **97.6** |
| | LSUN (C) | 69.4 / **76.0** | 75.7 / **79.9** | 84.3 / **87.4** | 89.3 / **92.8** |
| | Imagenet (R) | 70.8 / **76.6** | 80.2 / **80.2** | 95.0 / **95.7** | 95.4 / **95.5** |
| | Imagenet (C) | 72.5 / **78.2** | 78.7 / **80.2** | 92.1 / **93.6** | 93.2 / **93.6** |
| | Uniform | 81.7 / **93.5** | 96.7 / **96.8** | 100 / **100** | 100 / **100** |
| | Gaussian | 60.5 / **83.7** | 81.7 / **92.2** | 100 / **100** | 100 / **100** |
| | SVHN | 73.2 / **82.9** | 88.1 / **89.0** | 89.5 / **92.6** | 91.8 / **92.7** |
| CIFAR-10 | iSUN | 85.0 / **90.4** | 86.9 / **92.0** | 98.2 / **99.1** | 98.8 / **99.0** |
| | LSUN (R) | 85.3 / **90.8** | 87.1 / **92.4** | 98.7 / **99.3** | 99.1 / **99.2** |
| | LSUN (C) | 86.2 / **90.0** | 87.2 / **88.7** | 92.8 / **95.6** | 95.7 / **97.2** |
| | Imagenet (R) | 84.9 / **89.0** | 86.3 / **90.4** | 97.9 / **98.8** | 98.5 / **98.7** |
| | Imagenet (C) | 85.3 / **89.4** | 86.3 / **89.9** | 96.3 / **97.7** | 97.5 / **98.3** |
| | Uniform | 93.5 / **98.8** | 99.3 / **99.9** | 100 / **100** | 100 / **100** |
| | Gaussian | 95.5 / **99.7** | 99.8 / **100** | 100 / **100** | 100 / **100** |
| | SVHN | 85.1 / **90.3** | 89.1 / **93.0** | 96.8 / **98.1** | 98.1 / **98.4** |
| SVHN | iSUN | 89.7 / **92.8** | 89.2 / **93.5** | 98.2 / **99.1** | 99.7 / **99.9** |
| | LSUN (R) | 88.9 / **92.1** | 88.2 / **92.7** | 98.6 / **99.2** | 99.8 / **99.9** |
| | LSUN (C) | 89.7 / **92.2** | 89.2 / **92.2** | 94.8 / **97.3** | 98.0 / **98.9** |
| | Imagenet (R) | 90.4 / **93.4** | 90.0 / **94.2** | 98.0 / **99.1** | 99.5 / **99.8** |
| | Imagenet (C) | 91.0 / **93.3** | 90.6 / **93.8** | 97.1 / **98.7** | 99.2 / **99.6** |
| | Uniform | 92.9 / **95.7** | 92.3 / **97.4** | 99.9 / **100** | 100 / **100** |
| | Gaussian | 92.9 / **95.4** | 93.0 / **97.5** | 100 / **100** | 100 / **100** |
| | CIFAR-10 | 90.0 / **93.1** | 89.4 / **93.4** | 92.2 / **96.2** | 96.9 / **98.5** |
| | CIFAR-100 | 89.6 / **92.5** | 89.0 / **93.1** | 92.4 / **96.1** | 97.0 / **98.5** |

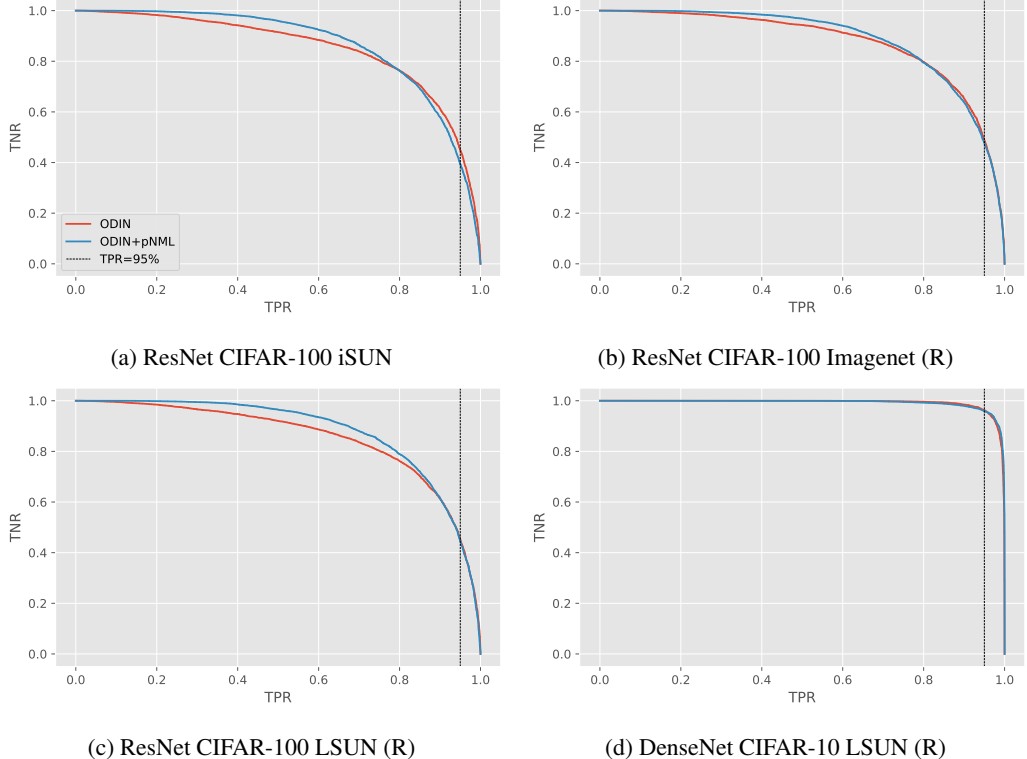

(a) ResNet CIFAR-100 iSUN

(b) ResNet CIFAR-100 Imagenet (R)

(c) ResNet CIFAR-100 LSUN (R)

(d) DenseNet CIFAR-10 LSUN (R)

Figure 4: The TNR as a function of the TPR of IND-OOD sets for which the ODIN method is better than the pNML at TPR of 95%.