# OpenReview forum: "Single Layer Predictive Normalized Maximum Likelihood for Out-of-Distribution Detection"
_NeurIPS.cc/2021/Conference — NeurIPS 2021 Poster_

### Official Review · Reviewer_t8Jy · 2021-07-12

**Rating:** 7
**Confidence:** 4

**Summary:**

This paper derives an explicit expression of the predictive normalized maximum likelihood (pNML) learner a single layer neural network probability assignment and regret (Eq. 7). This derived pNML regret expression is applied for the OOD detection task from pre-trained NN models. The authors also connect their pNML framework for single layer NN with soft-max activations and demonstrate that it can be applied for DNN models to obtain consistent improvements for the OOD detection performance.


----------------------
After rebuttal:- Overall, I am satisfied with the paper. I am increasing my final score to 7 for the paper.

Few additional comments:-
"... more closely related to the domain of adversarial attacks":
I am not sure about this comment. Typically adversarial examples deal with minor perturbations. However in this case, I was talking about "any input sample that is far away from the training distributions". I think that should be part of out-of-distribution.

"synthetic dataset":- Please label Feature 1 and Feature 2 values in the plot.




**Limitations And Societal Impact:**

I did not find any direct limitations and potential negative societal impact of this work.

**Main Review:**

Overall, the paper is well-written and the idea of pNML is nicely motivated. I have a few questions and suggestions as follows:

1. Related works:-

[Line 81] Lee et al. (2018) work with pre-trained models. They do not require to retrain their model. Also, there are a number of models that explicitly retrain their OOD detection models using additional OOD training datasets (e.g. [1-4]).
In contrast, frameworks such as ODIN, Mahalanobis, etc. do not retrain the model. Instead, they fine-tune/compute some hyper-parameters during inference.
Also, some other Bayesian-NN models do not use any additional data [5-7]. Please clearly mention these related works and the differences in Section 2.


2. compute the inverse of the data matrix (Line 216):-

It seems that you need to compute the inverse of the data matrix to obtain the regret score (Eq. 20 and 21). Does it lead to a bottleneck for scalability when the training size becomes very large? Please comment.


3. "If the test sample is far from the decision boundary, the ERM assigns to one of the labels probability 1. In this case, the regret is 0 no matter in which subspace the test vector lies." (Line 195-196)

Typically, any input sample that is far away from the training distributions is expected to produce high uncertainty scores and should be detected as OOD. Such data points can be also far away from the decision boundary. For example, see Figure 1 in [7]. However, this seems to be contradicting the idea of OOD. Please comment.

(Continued from the previous question) The analysis shown in Figure 1 is useful to understand the behavior of the proposed framework.
However, according to your claim, as we move top-left or bottom-right, we should never get an OOD data point with high regret. This does not seem to be right.

So, can you please expand the rectangular space and show the values of the regret as we move far away from the original training data points?
Also, can you please also consider an overlapping synthetic dataset and demonstrate the regret scores? (e.g., as in [3] Figure 5).


[1] Deep Anomaly Detection with Outlier Exposure, ICLR 2019

[2] Predictive Uncertainty Estimation via Prior Networks, NeurIPS 2018

[3] Towards Maximizing the Representation Gap between In-Domain & Out-of-Distribution Examples, NeurIPS 2020

[4] Self-Supervised Learning for Generalizable Out-of-Distribution Detection, AAAI 2020

[5] Simple and scalable predictive uncertainty estimation using deep ensembles, NeurIPS 2017

[6] Dropout as a bayesian approximation: Representing model uncertainty in deep learning, ICML 2016

[7] Uncertainty Estimation Using a Single Deep Deterministic Neural Network, ICML 2020


**Time Spent Reviewing:**

~10 hours.

---

> ### Author Response · Authors · 2021-08-10
> **Authors' response**
>
> We thank the reviewer for their careful review and positive comment on the paper’s writing and the motivation of the pNML. We address comments raised in the review,  in order.
> \
> "Related works: [Line 81] Lee et al. (2018) work with..."
>
> Shall be done.
>
> \
> “It seems that you need to compute the inverse of the data matrix to obtain the regret score (Eq. 20 and 21). Does it lead to a bottleneck for scalability when the training size becomes very large?”
>
> The embedding sizes of DensNet-100 and Resnet-18 are 342 and 512 respectively. For computing the inverse of the data matrix one needs to calculate the empirical correlation matrix of the training data $X_N^\top X_N$ which is a $M \times M$ matrix. This correlation matrix for training set size $N=1E6$ and embedding size $M=512$ takes 2.4 sec . The inverse of $M \times M$ is executed once per pretrained model: For $M=512$, the inverse takes 0.31 sec.
>
> This evaluation was executed on Intel(R) Xeon(R) Gold 6138 CPU @ 2.00GHz. For reference, the ImageNet training set size is $N=1.2E6$. Given these results we claim that computing the inverse is negligible compared to the model training in both runtime and resources. We will provide these details in the paper.
>
> \
> “Typically, any input sample that is far away from the training distributions is expected to produce high uncertainty scores and should be detected as OOD. Such data points can be also far away from the decision boundary.”
>
> In this work, we derived the OOD detection method based on a theoretical motivation. We agree that OOD samples can emerge in areas where the ERM learner is very confident. We believe, however, that areas where the predictor is very confident in its prediction but is still wrong, are more closely related to the domain of adversarial attacks. We are indeed looking into that direction for our future work.
>
> Our measure of confidence is the distance from a genie, a learner that knows the true label of the test sample but is restricted to a given model class. In areas where the ERM is highly confident  (i.e. the maximum probability of the prediction is 1.0) even the genie will make the wrong prediction since a single sample in these areas will not change its overall probability assignment. In the experiment section, Sec. 5, pp. 7-8, we show that it is not common for OOD samples to be in areas where the learner is confident of its prediction: The baseline method which detects OOD based on the max probability of the classifier manages to detect OOD.
>
> \
> “Can you please also consider an overlapping synthetic dataset and demonstrate the regret scores”
>
> The figure of overlapping data is shows here https://imgur.com/a/sGQpabF .
>
> The ERM probability is lower than 1.0 for most values of the test features. The pNML regret is small in the area of the data (also in the mixed label area). The regret is large in areas where there is no data, including the image periphery.
> We include this figure in the final paper since it provides the full picture of the pNML regret characteristic.

---

### Official Review · Reviewer_43yn · 2021-07-16

**Rating:** 6
**Confidence:** 3

**Summary:**

This paper propose using Predictive Normalized Maximum Likelihood (pNML) as the score for deep OOD prediction. The pMNL score is derived by computing the worst-case regret with respect to a oracle under MSE loss and with the model class constrained to a single-layer NN. Authors tested the effectiveness of pNML score for OOD detection by building it on top of various other works (Baseline, ODIN, Gram, OECC), showing consistent improvement over the baseline approaches.

**Limitations And Societal Impact:**

Yes

**Main Review:**

**Originality & Significance**:
Decent. The author derived a closed-form expression for pNML regret and empirically illustrated consistent improvement over recent approaches. The proposal appears to be simple and effective, and should be of interest to the NeurIPS community.

**Quality**: The derivation appears to be technically sound. The experiment setup used is standard among the OOD community.

**Clarity**
The paper is well structured and clearly written. However, the description of experiment setup can be improved. Specifically, please provide sufficient detail about how pNML is built on top of each methods (e.g., ODIN+pNML, Gram+pNML, etc). Specifically, it is not clear to me how the score for Gram+pNML is generated, since this method does not seems to modify the base architecture. If author is building on Gram method by simply computing pNML score by using the last layer of the architecture used by Gram, should the result be same as Baseline+pNML?



**Time Spent Reviewing:**

2

---

> ### Author Response · Authors · 2021-08-10
> **Authors' response**
>
> We thank the reviewer for considering this paper original and significant.
>
> Regarding your question on our experimental setup, we describe belew how we build the pNML on top of each method and we will add it to the paper (similar question also raised by reviewer Bp5r). Also, we provided the full code in the appendix and we will open source it.
>
> Briefly: \
> __Baseline (Hendrycks and Gimpel, 2017):__ A naive method that assigns the confidence score based on the maximum probability of a trained DNN. Input with a low maximum probability is considered OOD. We termed our pNML regret based score with a regular pretrained model as "Baseline+pNML".
>
> __ODIN (Liang et al., 2018):__ This method manipulates the input by perturbing it with the loss gradient. The score is the DNN's output maximum probability when feeding it with the perturbed input. We use the embeddings of the perturbed test image in the pNML regret calculation and name it "ODIN+pNML"'.
>
> __Gram (Sastry and Oore, 2020):__ This approach detects OOD samples based on feature representations obtained in intermediate layers of a pretrained DNN. Given a test sample, the Gram matrices of the test sample features obtained at different channels of the same layer are compared with those of the training samples known to belong to the estimated class of the test sample.
> Denote $\delta$ as the score of this method, samples with large $\delta$ are considered OOD. Samples with large $\delta$ are considered OOD. We utilize the Gram method by scaling the norm of the test sample embeddings, i.e., the regret score (eq. 20) becomes
> $\Gamma =\log \sum_{i=1}^{C} \frac{p_i}{p_i + p_i^{\delta x^\top g}\left(1 - p_i\right)}$. \
> We name this approach "Gram+pNML".
>
> __OECC (Papadopoulos et al., 2021):__  In this method there is a use of OOD samples during the training phase. A regularization term is added to the loss function such that the model produces a uniform distribution for OOD samples. We use the trained model by this method as the feature extractor and name it "OECC+pNML".
>
> __Energy (Liu et al., 2020):__ We use the variant of this approach that trains the model using OOD data. The loss term is a combination of the cross entropy loss with a regularization term that penalizes IND samples that produce high energy while encourages OOD samples to have high energy. See [5] eq. 8.
> We used the trained model by this approach as the feature extractor to produce our pNML score and name it "Energy+pNML".

---

> > ### Comment · Reviewer_43yn · 2021-08-14
> > **Thanks for the response!**
> >
> > Thanks authors for answering my questions! I still vote for acceptance and maintain my original score of 6.

---

### Official Review · Reviewer_A4Wx · 2021-07-16

**Rating:** 6
**Confidence:** 3

**Summary:**

The paper analytically derived the pNML regret for the single layer neural network, and apply it to any pre-trained NN by treating the layers as feature extractor and the last layer as a single layer NN for classification. Experimental results show that it can significantly improve the OOD detection performances of pre-trained classifiers.

**Limitations And Societal Impact:**

yes

**Main Review:**

The paper is well written, with a good balance between theoretical analysis and experimental evaluation. Although the analysis is a simplified version as it only considers single layer case, but it is good that the same method can be applied to any classifiers by looking at the last layer. The empirical results arecomprehensive and impressive, as it shows significant improvements over different models and datasets.

My only concern is that, it would be nicer to include experiments on larger dataset instead of only focuses on 32 resolution images. Recently, some paper on OOD detection include results on both small and large images (such as ImageNet-30 dataset, a subset of ImageNet), and this would make the results more convincing.

In general, I think the paper is theoretically sound and the results are convicing, therefore I vote for accept.

**Time Spent Reviewing:**

1.5

---

> ### Author Response · Authors · 2021-08-10
> **Authors' response**
>
> We thank the reviewer for their time and for remarking that our paper strikes a good balance between theoretical analysis and experimental evaluation.
>
> Following the reviewer’s suggestion to experiment with larger resolution images, we performed tests on Imagenet-30 [1]. We use the pre-trained PyTorch Resnet-18 and ResNet-101 models trained on ImageNet. We utilize the Imanget-30 training set for computing $X_N^\top X_N$ and the Imanget-30 test set as the IND set. All images were resized to 254x254 pixels. We compare the result to the Baseline in the following tables.
>
> ResNet-18 AUROC:
>
> | OOD           | Baseline     | pNML  |
> | -----------------     | :----------------:     | :---------: |
> | iSUN          | 95.58        | 99.74  |
> | LSUN (R)      | 95.51        | 99.72  |
> | LSUN (C)      | 96.89        | 99.77  |
> | Uniform       | 99.35        | 99.99  |
> | Gaussian      | 98.78        | 100     |
> | SVHN          | 99.18        | 99.99  |
> | CIFAR-10      | 89.99        | 99.79  |
> | CIFAR-100     | 92.15        | 99.74  |
>
> ResNet-101 AUROC:
>
> | OOD           | Baseline     | pNML  |
> | ----------------     | :----------------:     | :-------:       |
> | iSUN          | 96.26        | 99.54   |
> | LSUN (R)      | 95.77        | 99.43  |
> | LSUN (C)      | 98           | 99.86  |
> | Uniform       | 99.7         | 100     |
> | Gaussian      | 98.61        | 100     |
> | SVHN          | 98.94        | 99.98  |
> | CIFAR-10      | 91.24        | 99.47  |
> | CIFAR-100     | 93.39        | 99.58  |
>
> These tables show that we improve the AUROC of the baseline method by up to 9.8% and 8.23% for ResNet-18 and ResNet-101 models respectively. The result shows that the pNML regret is beneficial to OOD detection for large images (254x254 pixels).
> These new results shall be included in the paper.
>
> [1] D. Hendrycks, M. Mazeika, S. Kadavath, and D. Song. Using self-supervised learning can improve model robustness and uncertainty. In Advances in Neural Information Processing Systems, 2019

---

> ### Comment · Reviewer_A4Wx · 2021-08-27
> **Thanks for the reply**
>
> Nice to see the results on larger images. I will keep my score.

---

### Official Review · Reviewer_Bp5r · 2021-07-23

**Rating:** 6
**Confidence:** 4

**Summary:**

This paper generalizes the *predictive normalized maximum likelihood (pNML)* learner to single-layer neural network (linear regression with softmax) and show its application to detect out of distribution test samples for neural networks. The pNML learner can output a regret value (defined with respect to the learner that knows the actual test label) for each testing sample and this regret value is used for classifying the testing samples as out-of-distribution. The proposed method achieves strong empirical performance on image recognition tasks and can be further combined with existing OOD methods.

I find the proposed method to be novel and empirically strong. Therefore, I am leaning toward accepting. I think the authors have done a good job setting up the intuition and explaining the pNML learner. However, I wish the authors had done more analysis for the proposed method when combined with neural network.

**Ethical Concerns:**

I don't have ethical concerns with this submission

**Ethics Review Area:**

["I don’t know"]

**Limitations And Societal Impact:**

I do not see any immediate negative societal impact from this work

**Main Review:**

Pros

- the proposed method is novel
- the proposed method is empirically strong
- the submission is well-written and clear
- the proposed method doesn't not introduce high computational overhead unlike methods that require retraining
- the proposed method does not require additional data

Cons/Questions

- Can the authors clarify how pNML is used together with other OOD detection methods in Table 1 and Table 2?
- Fundamentally the success of this method relies on the quality of the feature extractor. After all, a feature extractor that maps any input into the same output is hopeless for OOD detection. I think the authors should acknowledge this fact in their discussion in section 4.2
- I wonder if there are more analyses the authors can do to support their empirical evaluation. For example, qualitatively, how is this proposed method different from existing measures such as the Baseline method?

**Time Spent Reviewing:**

6

---

> ### Author Response · Authors · 2021-08-10
> **Authors' response**
>
> We thank the reviewer for their time and kind words on the novelty of our work and the advantages of using the proposed method for OOD detection. We address the reviewer’s questions as follows.
>
> \
> “Can the authors clarify how pNML is used together with other OOD detection methods...”
>
> Certainly, and will also be clarified in the paper.  Briefly: \
> __Baseline (Hendrycks and Gimpel, 2017):__ A naive method that assigns the confidence score based on the maximum probability of a trained DNN. Input with a low maximum probability is considered OOD. We termed our pNML regret based score with a regular pretrained model as "Baseline+pNML".
>
> __ODIN (Liang et al., 2018):__ This method manipulates the input by perturbing it with the loss gradient. The score is the DNN's output maximum probability when feeding it with the perturbed input. We use the embeddings of the perturbed test in the pNML regret calculation and name it "ODIN+pNML".
>
> __Gram (Sastry and Oore, 2020):__ This approach detects OOD samples based on feature representations obtained in intermediate layers of a pretrained DNN. Given a test sample, the Gram matrices of the test sample features obtained at different channels of the same layer are compared with those of the training samples known to belong to the estimated class of the test sample.
> Denote $\delta$ as the score of this method, samples with large $\delta$ are considered OOD.
> We utilize the Gram method by scaling the norm of the test sample embeddings, i.e., the regret score (eq. 20) becomes
> $ \Gamma =\log \sum_{i=1}^{C} \frac{p_i}{p_i + p_i^{\delta x^\top g}\left(1 - p_i\right)}$.
> We name this approach "Gram+pNML".
>
> __OECC (Papadopoulos et al., 2021):__  In this method there is a use of OOD samples during the training phase. A regularization term is added to the loss function such that the model produces a uniform distribution for OOD samples. We use the trained model by this method as the feature extractor and name it “OECC+pNML”.
>
> __Energy (Liu et al., 2020):__ We use the variant of this approach that trains the model using OOD data. The loss term is a combination of the cross entropy loss with a regularization term that penalizes IND samples that produce high energy while encourages OOD samples to have high energy. See [5] eq. 8.
> We used the trained model by this approach as the feature extractor to produce our pNML score and name it “Energy+pNML”.
>
> \
> “Fundamentally the success of this method relies on the quality of the feature extractor."
>
> Correct. Importantly, our method is agnostic to the choice of a feature extractor. A poor one, such as the one described by the reviewer, will, therefore, indeed lead to poor results. At the same time, selecting feature extractors better than those used in our experiments, could potentially improve on our results. The reliance on the feature extractor can also be seen in Tables 1,2:  in the difference between Baseline+pNML and OECC+pNML. The OECC method trains the model with OOD samples such that the model produces uniform probability for OOD which makes the OECC model works better as a feature extractor than the Baseline. We will clarify it in the paper.
>
> \
> “I wonder if there are more analyses the authors can do to support their empirical evaluation.“
>
> In response to the reviewers’ suggestions, we performed the following experiments: a) Expanding the pNML empirically beyond 1 layer, b) Evaluating the pNML on the ImangeNet-30 set which uses 254x254 pixel images. We describe these new results below and we will add these new empirical evaluations to our paper.
>
> In addition, we also offer the information provided in Appendix C  which examines the spectrum of a real dataset  and Appendix D, which compares the pNML to the Gram method and shows that most of the pNML improvement is in assigning a high score to OOD samples while there is not much change in the rank of the IND ones.
> We welcome any additional suggestions the reviewer can provide and will gladly add those to the paper as well.
>
> __Expanding the pNML beyond 1 layer__ \
> We evaluate how the accuracy and log-loss depend on the number of layers that are fine-tuned in the pNML process. Notice that fine-tuning is executed for each test sample and for each possible value of the test label.
> We use ResNet-18 with CIFAR-10 and MNIST training sets. The initial training consisted of 100 epochs with a learning rate of 0.01 for the first 10 epochs, 0.01 for the next 30, 0.001 for the next 40, and 0.0001 for the rest. Next, We executed the pNML procedure. The fine-tuning phase consisted of 10 epochs with a learning rate of 0.0001, changing either the 1 last later, the 2 last layers or all the model layers.
> The results are shown in the following table.
>
>
> | Training set  |  Model         |  Acc      | Loss   | Worst-case logloss     |
> |------------------| -------------------    | :-------:     | :--------: | :-------------------------:     |
> | CIFAR-10     | ERM         | 0.920 | 0.203 |      0.431        |
> |                             | 1 layers     | 0.920 | 0.202 |      0.429        |
> |                             | 2 layers     | 0.921 | 0.173 |      0.361        |
> |                         | All layers       | 0.913 | 0.24   |      0.341        |
> | MNIST        | ERM         | 0.930 | 0.093 |     0.163        |
> |                    | 1 layers    | 0.931 | 0.090 |     0.159        |
> |                    | 2 layers    | 0.937 | 0.086 |     0.121        |
> |                  | All layers      | 0.937 | 0.085 |      0.115        |
>
>
> Training 2 layers seems to improve the performance of the model especially in the log-loss metric. However, when fine-tuning all model layers there is a degradation in performance.
> This degradation can be interpreted as an “insurance'' the pNML pays to protect against large loss: in general, when the model class increases, the regret increases, while a better fit to the data is obtained. The increased regret can be thought of as the cost insurance, to protect against large log-loss of unfit data. We can see in the table that while on the average the large model class performs poorer (as it pays the “insurance” anyway) it is better in the worst case. Notice that the pNML is the solution for the worst-case regret as described in eq. 6.
>
>
> __Evaluating the pNML on ImangeNet-30__ \
> We use the pre-trained PyTorch Resnet-18 and ResNet-101 models trained on ImageNet. We utilize the Imanget-30 [1] training set for computing $X_N^\top X_N$ and the Imanget-30 test set as the IND set. All images were resized to 254x254 pixels. We compare the result to the Baseline in the following tables.
>
> Resnet-18 AUROC:
>
> | OOD           | Baseline     | pNML  |
> | -----------------     | :----------------:     | :---------: |
> | iSUN          | 95.58        | 99.74  |
> | LSUN (R)      | 95.51        | 99.72  |
> | LSUN (C)      | 96.89        | 99.77  |
> | Uniform       | 99.35        | 99.99  |
> | Gaussian      | 98.78        | 100     |
> | SVHN          | 99.18        | 99.99  |
> | CIFAR-10      | 89.99        | 99.79  |
> | CIFAR-100     | 92.15        | 99.74  |
>
> ResNet-101 AUROC:
>
> | OOD           | Baseline     | pNML  |
> | ----------------     | :----------------:     | :-------:       |
> | iSUN          | 96.26        | 99.54   |
> | LSUN (R)      | 95.77        | 99.43  |
> | LSUN (C)      | 98           | 99.86  |
> | Uniform       | 99.7         | 100     |
> | Gaussian      | 98.61        | 100     |
> | SVHN          | 98.94        | 99.98  |
> | CIFAR-10      | 91.24        | 99.47  |
> | CIFAR-100     | 93.39        | 99.58  |
>
> These tables show that we improve the AUROC of the baseline method by up to 9.8% and 8.23% for ResNet-18 and ResNet-101 models respectively.
>
> [1] D. Hendrycks, M. Mazeika, S. Kadavath, and D. Song. Using self-supervised learning can improve model robustness and uncertainty. In Advances in Neural Information Processing Systems, 2019

---

### Official Review · Reviewer_fLaK · 2021-07-25

**Rating:** 6
**Confidence:** 3

**Summary:**

This paper studies out-of-distribution detection with single-layer neural networks. There are theoretical and experimental results - for theoretical contributions the authors derive the analytical form of the NML regret, and for experimental contributions the authors run their method on top of a pre-trained neural network, where they achieve superior performance in out-of-distribution data collection for the CIFAR and SVHN datasets.

**Limitations And Societal Impact:**

Yes, the authors discussed several limitations and potential negative societal impacts in the conclusion.

**Main Review:**

The paper is overall well written and well-motivated and the contributions are clear.

I have some concerns with the magnitude and significance of the contributions in the paper, but of course these are my own opinions and I would like to hear the authors opinions. The "single-layer" neural network discussed in the paper is really just a softmax linear classifier/multi-class logistic regression. In terms of writing, to avoid potential confusion for readers I believe it should just be referred to by a more standard name, such as logistic regression. The application for DNNs also only applies to training the final layer, which reduces the problem to logistic regression since the static network is just a fixed feature transform, in contrast to actually training the entire network. This aspect seems slightly swept under the rug, and maybe deserves more insight into the tradeoffs in the two potential approaches. The theory presented in the paper also seem more suited to be labeled corollaries rather than substantial theoretical contributions - they involve plugging the results derived in one paper (Zhuang et. al 2020) into the definitions for logistic regression and the NML distribution.

The experimental results seem solid, although I am not familiar with prior work in OOD detection, the metrics used, and what numbers constitute a strong result. Therefore, I will not make a strong claim in this area.

Even though the theoretical results do not hold in this case, have the authors compared fine-tuning the last layer of the neural network with NML, versus training the all layers of the network? This would also be an interesting ablation study because the full neural network would have higher regret due to the more expressive model class, and it would be interesting to see how this plays together with OOD detection. Is a more expressive class better or worse for this task?

**Time Spent Reviewing:**

1

---

> ### Author Response · Authors · 2021-08-10
> **Authors' response**
>
> Thank you for your time and kind words on the clarity of our motivation and contribution. We address your concerns by order.
>
> \
> “The single-layer neural network discussed in the paper is really just a softmax linear classifier”
>
> Good point. Importantly, we note that in the special case of binary classification, one can show that a single-layer neural network (which is often called softmax regression) reduces to logistic regression [1]. This shows that softmax regression is a generalization of logistic regression. We agree with your point that the name single-layer neural network might confuse the reader which is not our intention. We shall clarify this ambiguity in the paper.
>
> \
> “The theory presented in the paper also seem more suited to be labeled corollaries rather than substantial theoretical contributions”
>
> The reviewer’s comment is a subjective statement on our contributions. Such comments are generally hard to address. More objectively, we have shown the advantages of the approach described in our paper (demonstrated in our experiments Sec. 5 pp. 8-9) and provided a theoretical motivation in pp. 4 Theorem 1.  The suggested approach is novel and backed by theory and so we believe it is beyond “corollaries”.
>
> \
> “Have the authors compared fine-tuning the last layer of the neural network with NML, versus training all layers of the network?”
>
> In response to the reviewer's question, we conducted experiments exploring the tradeoff between executing the pNML with training a different number of layers. Our findings suggest that using more than one layer improves the performance especially for the worst case log-loss. We will share these new results in the paper.
>
> Specifically, we evaluate how the accuracy and log-loss depend on the number of layers that are fine-tuned in the pNML process. Notice that the fine-tuning is executed for each test sample and for each possible value of the test label.
> We use ResNet-18 with CIFAR-10 and MNIST training sets. The initial training consisted of 100 epochs with a learning rate of 0.01 for the first 10 epochs, 0.01 for the next 30, 0.001 for the next 40, and 0.0001 for the rest. Next, We executed the pNML procedure. The fine-tuning phase consisted of 10 epochs with a learning rate of 0.0001, changing either the last layer, the 2 last layers or all the model layers.
> The results are shown in the following table.
>
>
> | Training set  |  Model         |  Acc      | Loss   | Worst-case logloss     |
> |------------------|-------------------    | :-------:     | :--------: | :-------------------------:     |
> | CIFAR-10     | ERM         | 0.920 | 0.203 |      0.431        |
> |                             | 1 layers     | 0.920 | 0.202 |      0.429        |
> |                             | 2 layers     | 0.921 | 0.173 |      0.361        |
> |                         | All layers       | 0.913 | 0.24   |      0.341        |
> | MNIST        | ERM         | 0.930 | 0.093 |     0.163        |
> |                    | 1 layers    | 0.931 | 0.090 |     0.159        |
> |                    | 2 layers    | 0.937 | 0.086 |     0.121        |
> |                  | All layers      | 0.937 | 0.085 |      0.115        |
>
>
> Training 2 layers seems to improve the performance of the model especially in the log-loss metric. However, when fine-tuning all model layers there is a degradation in performance.
> This degradation can be interpreted as an “insurance'' the pNML pays to protect against large loss: in general, when the model class increases, the regret increases, while a better fit to the data is obtained. The increased regret can be thought of as the cost insurance, to protect against large log-loss of unfit data. We can see in the table that while on the average the large model class performs poorer (as it pays the “insurance” anyway) it is better in the worst case. Notice that the pNML is the solution for the worst-case regret as described in eq. 6.
>
> We believe that for over-parameterized models additional constraints should be imposed in order to constrain the maximal regret. For example the norm of the parameters should be equal to the ERM norm or constrain the number of fine-tuning epochs in a non-heuristic way. There is some previous work that explores these constraints, e.g Zhou and Levine (2020) and [2]. [2] will be added to the paper.
>
> [1] Wolfe, J., Jin, X., Bahr, T., & Holzer, N. (2017). Application of softmax regression and its validation for spectral-based land cover mapping. The International Archives of Photogrammetry, Remote Sensing and Spatial Information Sciences, 42, 455.
>
> [2] Koby Bibas, & Meir Feder (2021). Distribution Free Uncertainty for the Minimum Norm Solution of Over-parameterized Linear Regression. Workshop on Distribution-Free Uncertainty Quantification ICML 2021‏.

---

> > ### Comment · Reviewer_fLaK · 2021-08-17
> > **Follow-up**
> >
> > Thank you for the detailed response. After reading the rebuttal and the other reviews, I am inclined to accept the paper and raise my score to a 6.

---

### Decision · Program_Chairs · 2021-09-27

**Decision:**

Accept (Poster)

**Comment:**

The paper proposes to use predictive normalized maximum likelihood (pNML) for detecting out-of-distribution inputs.

Overall, the reviewers found it be a well-written paper. The idea of pNML for OOD detection is novel and the empirical results show consistent improvements over baseline. The authors did a good job of addressing major reviewer concerns. During the discussion phase, the consensus decision learned towards acceptance. I recommend acceptance and encourage the authors to address the remaining comments in the camera ready version.

Other suggestions to improve the final version:
- I'd encourage the authors to evaluate the technique on harder OOD pairs (e.g. https://arxiv.org/abs/2007.05566 define near-OOD pairs such as CIFAR-100 vs CIFAR-10) as it would be interesting to see how the method performs on more difficult benchmarks.
- Section 4: The idea of weighting Eigen directions with large Eigen values seems related to variants of Mahalanobis distance such as marginal Mahalanobis distance https://arxiv.org/abs/2003.00402 and Relative Mahalanobis distance https://arxiv.org/abs/2106.09022 It would be interesting to add a discussion and potentially compare pNML to these methods.